# Importance of missingness in baseline variables: A case study of the *All of Us* Research Program

**Robert M. Cronin**[1‡]*, **Xiaoke Feng**[2‡], **Lina Sulieman**[3], **Brandy Mapes**[4], **Shawn Garbett**[2], **Ashley Able**[4], **Ryan Hale**[4], **Mick P. Couper**[5,6,7], **Heather Sansbury**[8], **Brian K. Ahmedani**[9], **Qingxia Chen**[2,3]*

**1** Department of Internal Medicine, The Ohio State University, Columbus, Ohio, United States of America, **2** Department of Biostatistics, Vanderbilt University Medical Center, Nashville, Tennessee, United States of America, **3** Department of Biomedical Informatics, Vanderbilt University Medical Center, Nashville, Tennessee, United States of America, **4** Vanderbilt Institute for Clinical and Translational Research, Vanderbilt University Medical Center, Nashville, Tennessee, United States of America, **5** Survey Research Center, University of Michigan, Ann Arbor, Michigan, United States of America, **6** Department of Biostatistics, School of Public Health, University of Michigan, Ann Arbor, Michigan, United States of America, **7** Survey Research Center, Institute for Social Research, University of Michigan, Ann Arbor, Michigan, United States of America, **8** National Institutes of Health, Bethesda, Maryland, United States of America, **9** Center for Health Policy & Health Services Research, Henry Ford Health, Detroit, Michigan, United States of America

‡ RMC and XF are co-first authors on this work.
* robert.cronin@osumc.edu (RC); cindy.chen@vumc.org (QC)

**Data Availability Statement:** Data is owned by a third party, the All of Us Research Program. The data underlying this article were provided by the All of Us Research Program by permission that can be

## Abstract

### Objective

The *All of Us* Research Program collects data from multiple information sources, including health surveys, to build a national longitudinal research repository that researchers can use to advance precision medicine. Missing survey responses pose challenges to study conclusions. We describe missingness in *All of Us* baseline surveys.

### Study design and setting

We extracted survey responses between May 31, 2017, to September 30, 2020. Missing percentages for groups historically underrepresented in biomedical research were compared to represented groups. Associations of missing percentages with age, health literacy score, and survey completion date were evaluated. We used negative binomial regression to evaluate participant characteristics on the number of missed questions out of the total eligible questions for each participant.

### Results

The dataset analyzed contained data for 334,183 participants who submitted at least one baseline survey. Almost all (97.0%) of the participants completed all baseline surveys, and only 541 (0.2%) participants skipped all questions in at least one of the baseline surveys. The median skip rate was 5.0% of the questions, with an interquartile range (IQR) of 2.5% to 7.9%. Historically underrepresented groups were associated with higher missingness

sought by scientists and the public alike. The Researcher Workbench is a cloud-based platform where registered researchers can access Registered and Controlled Tier data, including the data presented here. Researchers/citizen scientists must verify their identity and complete the All of Us Research Program data access process to access the Researcher Workbench and Registered Tier data. Once this process is completed, the data will be made available to all persons. More information on data access can be found in the All of Us Research Hub (https://www.researchallofus.org/) as is the option to register for access. The authors did not have any special access privileges to this data that other researchers would not have.

**Funding:** This work was funded by the National Institutes of Health (https://allofus.nih.gov/) 5U2COD023196 (RMC, XF, LS, BMM, SG, AS, RJ, MPC, QC) and 3OT2OD026550 (BKA). (Additional support included the National Heart, Blood, and Lung Institute (https://www.nhlbi.nih.gov/) K23HL141447 (RMC). The funders had no role in study design, data collection and analysis, decision to publish, or preparation of the manuscript.

**Competing interests:** The authors have declared that no competing interests exist.

(incidence rate ratio (IRR) [95% CI]: 1.26 [1.25, 1.27] for Black/African American compared to White). Missing percentages were similar by survey completion date, participant age, and health literacy score. Skipping specific questions were associated with higher missingness (IRRs [95% CI]: 1.39 [1.38, 1.40] for skipping income, 1.92 [1.89, 1.95] for skipping education, 2.19 [2.09–2.30] for skipping sexual and gender questions).

## Conclusion

Surveys in the All of Us Research Program will form an essential component of the data researchers can use to perform their analyses. Missingness was low in *All of Us* baseline surveys, but group differences exist. Additional statistical methods and careful analysis of surveys could help mitigate challenges to the validity of conclusions.

## Introduction

Understanding the patterns of missing data is vital for any scientific research project. If data are incomplete, there are potential threats to the validity of conclusions that use those data [1]. Some of the most critical threats to validity include a loss of statistical power, data not missing completely at random, and how analyses and missingness are handled. A loss of statistical power can occur for complete case analyses where many of the target population are removed due to missing key variables. Excluding participants from analyses because of missing data can undermine the original goals of the study. If data are not missing completely at random, meaning if missing cases differ from non-missing cases on key outcomes or covariates, conclusions could be biased. This point is crucial in a large cohort study where significant effort is expended to recruit and retain diverse populations. Finally, different analyses could yield different results depending on which variables are included and what strategies are used to account for missing data on those variables [2].

Health surveys are traditional methods to collect data from participants in biomedical research. Since participants can choose what questions they want to respond to in health surveys, they may be especially susceptible to missing data [3]. Many articles describe potential biases of testing hypotheses with datasets having critical missingness [4]. In recent years, other articles have shown a decline in survey response rates, threatening the validity of conclusions drawn from these studies [5–9]. There are multiple ways to handle missing survey data in general [10], including multiple imputation [11], inverse probability weighting [12, 13], full likelihood [14], Fully Bayesian [15], or hybrid methods [16]. In surveys, missing data can occur when a subpopulation is not included in the survey's sampling frame (noncoverage), a sampled unit does not participate in the survey (total nonresponse), or because a responding sampled element fails to provide acceptable responses to one or more of the survey items (item nonresponse) [4]. Various methods have been developed to compensate for missing survey data in a generally purposeful way to mitigate the effect on estimates. Weighting adjustments are often used to compensate for noncoverage and total nonresponse. Imputation methods that assign values for missing responses compensate for item nonresponses. To make the best use of these methods, it is essential to first understand the levels and patterns of missingness within health surveys.

The *All of Us* Research Program, hereafter referred to as *All of Us*, has set out to collect information from over 1 million participants of diverse backgrounds historically

underrepresented in biomedical research to advance the science of precision medicine [17, 18]. The program collects data from participants through multiple sources, including electronic health records, digital health technology, biospecimens, and health surveys. These health surveys can augment and validate the information about participants from other sources, thereby helping researchers answer crucial biomedical research questions of precision medicine. Populations of diverse backgrounds that have been historically underrepresented in biomedical research may pose additional challenges in missingness from health surveys. It is anticipated that *All of Us* data will be heavily used by scientists worldwide [17]. Understanding the missingness in such an extensive program of participants usually underrepresented in biomedical research is of utmost importance.

*All of Us* created and launched seven surveys, three of which are available to participants when a participant initially enrolls in the program and are referred to here as baseline surveys. Participants will continue to receive surveys throughout the life of the program. The data from these surveys are currently available to researchers (https://www.researchallofus.org/); however, there is a gap in our understanding of missing data in these baseline surveys. By understanding what data are missing from the *All of Us* health surveys, why it is missing, and how to overcome missingness, scientific researchers can understand limitations and best address their research questions using this data resource.

The objective of this project was to use *All of Us* as a case study to demonstrate an approach to evaluate missingness of surveys and identify characteristics that are associated with missingness in a large epidemiological cohort. In particular, we studied if the demographical variables that define the historically underrepresented groups in biomedical research, enrollment date, and health literacy were associated with an increased risk of missingness in the remaining survey questions of the baseline surveys.

## Methods

### Overview

The initial three survey modules released at baseline were 1) The Basics, which covered basic demographic, socioeconomic, and health insurance questions; 2) Overall Health, which included the brief health literacy scale [19, 20], the overall health PROMIS scale [21], and questions important for collecting biospecimens, such as transplant and travel history; and 3) Lifestyle, which included questions about smoking, alcohol, and illicit drug use. The development of these surveys is described elsewhere [22]. These surveys contained branching logic, which was used to ensure that specific questions, often referred to as "child" or follow-up questions, were presented to participants based on selecting only a relevant previous question response. For example, if a participant has never had at least one drink of any kind of alcohol in their lifetime, they would not be asked questions about how often and how much they drank. Our analyses only included questions that the participant saw and did not respond to as missing. We excluded questions that the participant did not see from our analyses. All the potential questions and branching logic are available at: https://www.researchallofus.org/data-tools/survey-explorer/. Of the questions participants have seen, a participant can skip any question and progress to the next one. Once a participant completed a survey, the data were sent to a raw data repository at the *All of Us* Data Research Center at Vanderbilt University Medical Center. We extracted the survey responses from May 31, 2017, to September 30, 2020. All data presented were stripped of identifiable information. The Institutional Review Boards of the All of Us Research Program approved all study procedures and informed consent was obtained from all participants.

## Missingness analysis

Missingness can be evaluated by completing an entire survey or by specific questions within one survey. Missing data can be examined at the level of the participant, such as one participant not answering a set of questions, or by the item, such as a set of participants skipping one question. In this manuscript, we evaluated the missingness by the participant level because this allowed us to understand the pattern of missingness by participants' characteristics. We observed three types of missingness in this project: (a) missingness or no submission of an entire survey; (b) survey submission without answering any questions [23, 24]; and (c) item nonresponse, where specific but not all questions were skipped within a survey. We defined item nonresponse as when the participant saw the question and they did not respond to the question. Some questions also had explicit "Prefer not to answer" or "Don't know" options. Participants who responded with one of these options were not counted as missing in the primary analysis but were analyzed in a sensitivity analysis described below.

We reported the count and percentage of participants who did not submit each of the entire survey modules or skipped all the questions in a survey. For the participants who answered at least one question, we defined the missing percentage as the ratio of missing items to the number of corresponding branching-logic-based eligible questions. The demographic questions defining the historically underrepresented in biomedical research and health literacy questions were considered as explanatory variables and excluded from the missing percentage calculation. The missing percentages for various underrepresented groups were compared to represented groups using Wilcoxon rank-sum or Kruskal Wallis tests. Associations of missing percentages with age at enrollment, health literacy score, and enrollment date were evaluated using Spearman correlation coefficients. We performed a negative binomial regression to evaluate the impact of participant characteristics on the percentage of missingness of a participant based on the number of eligible questions for a participant allowing for overdispersion. The independent variables included race and ethnicity, age, education attainment, household income, sexual and gender minority, geography (non-urban versus urban status), health literacy score, and enrollment since *All of Us* initiation, which we defined as the number of weeks since *All of Us* started (May 2017) to the participant's enrollment date. The three continuous variables of age, and enrollment since *All of Us* initiation were modeled using a five-knot natural cubic spline to allow for a nonlinear association. As race/ethnicity was associated with an increased risk of missingness, we also investigated if the effect was moderated by education, age, and sex/gender by including the two-way interaction terms in the model.

A health literacy score was defined as the summation of three individual questions of the brief health literacy scale in the Overall Health survey. For the participants missing one or more of the three individual questions, multiple imputation was applied to individual questions using the *mice* package in R [25]. Six complete datasets were generated from the multiple imputation models [26] and analyzed using the negative binomial regression method described above. The health literacy scale had about 6% missingness. Therefore, we used six imputations to allow for a <1% efficiency loss [26, 27]. Estimates and standard errors for regression coefficients across the six datasets were combined into single estimates by averaging and using standard errors with Rubin's rules [27]. We performed the following additional sensitivity analyses: A) evaluated missingness using a negative binomial regression with complete case analysis, which removed the participants with missing values on any variables included in the model; B) applied multiple imputation on the total health literacy score instead of individual questions and repeated the same analysis with this imputed health literacy score; C) for participants who only missed one health literacy score questions, we used the average of the other scores to impute the total health literacy score then repeated two analyses described

above; D) we also counted "Prefer not to answer" or "Don't know" as missing in defining the missing percentage and repeated the primary analysis as an additional sensitivity analysis. Multiple imputation was applied to individual health literacy score questions, and a negative binomial regression was performed on the imputed dataset.

All analyses were performed using the R Programming Language 3.3.0 [25]. We considered P-values less than 0.05 a statistically significant difference. With the large sample size, 95% confidence intervals (CI) were reported.

## Results

### Descriptive analysis

The program had 334,183 participants who submitted at least The Basics baseline survey (the first baseline survey available for completion) between May 31, 2017, and September 30, 2020. Among those 334,183 participants, all three baseline surveys (The Basics, Overall Health, or Lifestyle) were completed by 323,693 (97.0%) participants, and only 541 (0.2%) participants skipped all questions in at least one of the three surveys. A subset of participants, 36,077 (10.8%), answered every eligible question. A vast majority, 250,304 (74.9%), skipped fewer than 10% of the questions, while very few, 522 (0.2%), skipped more than half of the questions. The median skip rate was 5.0% of the questions, with an interquartile range (IQR) of 2.5% to 7.9%.

### Different population characteristics had different levels of missingness

Compared to the mean, participants who skipped the educational attainment, race and ethnicity, household income, or sexual and gender questions skipped significantly more additional questions (Fig 1a). For example, 5741 participants (1.7%) who did not answer a race/ethnicity skipped 13% of the remaining questions of the baseline surveys, while the average skip rate was about 5% of the questions. Participants from specific populations historically underrepresented in biomedical research skipped more than the average, including those with less than high school education, Black or African American, and Latino or Spanish participants. Other underrepresented groups, such as sexual and gender minorities, rural geography, and older ages, had slightly lower missingness than the mean.

### The participant missing percentage was similar over time, participant ages, and participant health literacy score

The participant missing percentage has been relatively stable in *All of Us* thus far (Fig 1b). There were slight deviations before the national launch of the program in May 2018 and after the start of the COVID-19 pandemic in March 2020. The participant missing percentage was almost constant across different ages (Fig 1c). Between ages 68–78, the missing percentage was slightly decreased but stabilized after 78 years of age. The health literacy score ranged between 3 and 15, with higher scores indicating higher subjective health literacy. In Fig 1d, we only included participants who answered all three health literacy questions. The participant missing percentage did not change much as the health literacy score increased.

### Multivariable analysis

In the negative binomial regression analysis (Fig 2a–2d), when holding the other variables constant in the model, participants who skipped household income, race and ethnicity, educational attainment, and sexual and gender questions were more likely to have a higher overall missingness rate compared to those who didn't skip those questions (incidence rate ratios

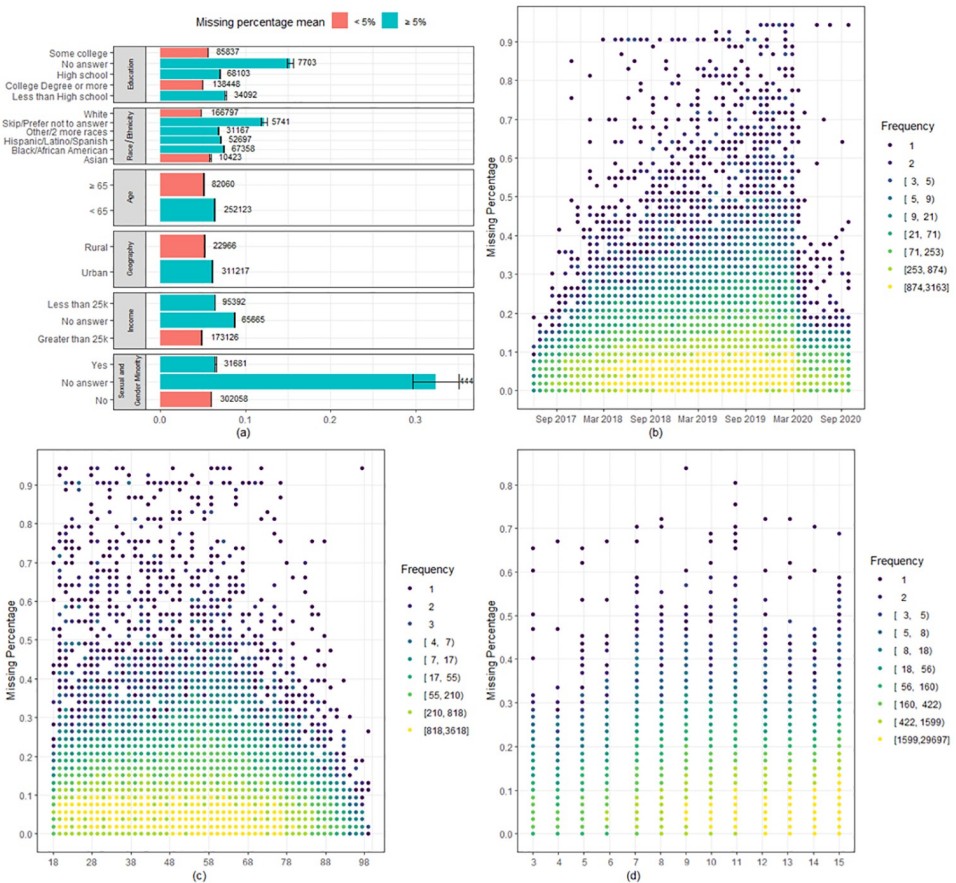

**Fig 1. Missingness by groups underrepresented in biomedical research, consent date, age, and health literacy scores.** (A) Missing percentage mean (B) Consent date (C) Age (D) Health literacy score.

(IRRs) [95% CI]: 1.39 [1.38, 1.40] for skipping income, 1.69 [1.66, 1.72] for skipping race and ethnicity, 1.92 [1.89, 1.95] for skipping education, 2.19 [2.09–2.30] for skipping sexual and gender questions). Participants from rural geography had a lower incident rate for missing questions than urban geography (IRR: 0.93, 95% CI: [0.92, 0.94]). Except for geography, underrepresented groups had higher incident rates compared with represented groups, especially for racial and ethnic minority groups (IRRs [95% CI]: 1.26 [1.25,1.27] for Black/African; 1.15 [1.14,1.16] for Hispanic/Latino; 1.22 [1.21,1.23] for other race or a combination of two or more races, all compared to White) and lower educational attainment (IRRs [95% CI]: 1.14 [1.13,1.15] for less than high school; 1.11 [1.10, 1.12] for high school, all compared to a college degree). The IRR increased before age 35 and after 65 but decreased between 35 and 65 (Fig 2b). In addition, participants with health literacy scores between 6 and 10 had a higher IRR than those who had higher or lower scores (Fig 2c). Participants who enrolled near the beginning of the *All of Us* program or enrolled more recently had lower IRRs than those enrolled in the middle period (Fig 2d). The results were similar in the sensitivity analyses (see the supplemental document, S1–S5 Figs). Some race/ethnicity interactions with sex/gender, age, and education were significant and demonstrated differential race/ethnicity effects moderated other baseline variables (see the supplemental document, S6 Fig). However, the most significant IRRs were among the groups with missingness in the baseline variables.

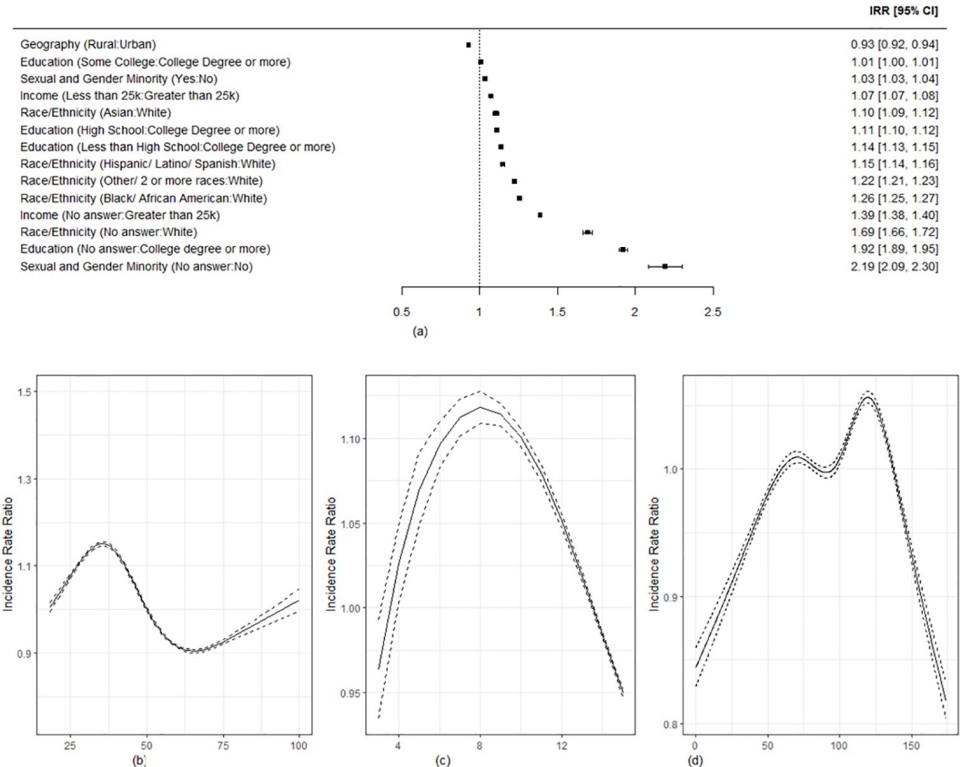

**Fig 2. Negative binomial analysis of missingness.** (A) Incidence Rate Ratio (B) Age (Years) (C) Health literacy score (D) Enrollment since All of Us initiation (in weeks).

## Discussion

For All of Us, the three baseline modules studied in this article were required for all participants and contained the information that could be commonly used in the All of Us research program. In this manuscript, we described the completeness of these survey data and identified some key baseline variables associated with the overall missingness of the rest of the variables from the three baseline survey modules. The fact that those often not included in biomedical research have higher rates of missingness is a potential red flag for studies, such as All of Us, that make special efforts to include such populations. The missingness of these background variables of race, sex and gender identity, education, time of enrollment, and geography, could contribute to non-random missingness in surveys. If these populations fail to answer other questions or drop out altogether, any complete case analysis will mean that these losses undermine the efforts to include such groups.

Very few participants only answered one of the first three surveys. The low level of nonresponse to these surveys is promising, and using all of the surveys for analyses appears to be a reasonable approach. Almost all participants who also started a survey did not simply click through without responding to any questions. A vast majority (74.9%) skipped less than 10% of the questions, while very few (0.2%) skipped more than half of the questions. While this number is low, researchers will need to be cautious in reviewing whether their population of interest may have "completed" surveys yet not have usable data for a portion of their population. Removing these participants from analyses may be a reasonable approach due to their rarity and a large amount of item missingness. Due to a large number of participants

answering most questions, researchers should be able to pursue hypotheses with the data without concern for bias.

Participants with different characteristics had different levels of missingness. Participants who skip a few of the baseline questions, such as race and ethnicity, sex and gender, educational attainment, and household income, are more likely to skip other questions. These participants had the highest missingness rates compared to participants who answered these questions. Also, these indicators were some of the most substantial risk factors for missingness in our regression models. These data suggest that participants unwilling to provide critical demographic details are also less willing to answer other survey questions. Participants from certain historically underrepresented populations skipped more than the mean, including those with less than a high school education and those who identify as Black or African American and Latino or Spanish. While only slightly lower missing percentages, certain underrepresented groups had lower missingness than the mean, such as sexual and gender minorities and those residing in rural geographic areas. Other characteristics did not differ in missingness rates, such as consent date, health literacy score, and age. Understanding the missingness of certain sociodemographic populations to understand the potential missingness of other questions is crucial.

Sensitivity analysis demonstrated similar results as the primary analysis. Multiple imputation demonstrated similar results as complete case analysis and average score imputation for the brief health literacy scale. A vast majority of the participants completed all three brief health literacy scale items. Our approach for this scale could be used for other scales in the *All of Us* dataset, such as the PROMIS overall health scale, to mitigate biases in the data. Also, considering "don't know" and "prefer not to answer" as missing values did not alter our results. However, researchers need to be cautious as certain questions, such as income, could have higher rates of missingness if these options are considered missing.

This study had several limitations. First, this is a snapshot of the data as of September 2020. While the missingness may change over time, we noted that the missingness has not historically changed a large amount. Second, we did not review additional surveys that were completed after baseline. Reviewing additional surveys is an area of future work to help understand the missingness of all survey data. Third, we did not review other data sources in the program, such as electronic health records. Other data sources within *All of Us* may augment missingness in surveys. Finally, additional variables may be important for missingness, such as having enrollment staff help participants with questions and responses they may not understand. Obtaining and evaluating these additional variables could help researchers understand the causes of missingness.

This analysis will help researchers of the *All of Us* data understand and assess data missingness. This manuscript serves as a complementary follow-up to the initial *All of Us* survey development manuscript [22], detailing quality assessment efforts routinely undertaken by the *All of Us* Data and Research Center, in collaboration with program partners, to understand the *All of Us* survey data composition and provide recommendations to researchers interested in applying similar checks of their data. This work will be put into the *All of Us* Researcher Workbench as featured notebooks and educational documentation that can be used by researchers in evaluating and understanding missing data as more data continues to come into the program.

The analyses presented in this manuscript can be used by researchers for the All of Us survey data and survey data from other large epidemiological cohorts like the Million Veterans Program or UK Biobank to help reduce potential biases and account for them. Another key message is that identifying "leading indicators" of missingness (i.e., the predictors) could help survey designers to target strategies to reduce differential missingness. The fact that we found

a low missing percentage in the three baseline survey modules of the All of Us cohort offers some reassurance to substantive research but also points to the importance of adjusting for the critical variables associated with overall missingness on substantive analyses. Researchers must be cautious when using complete case analysis or assuming missingness at random in All of Us or other large epidemiological cohorts (e.g., UK biobank, Million Veterans Program).

## Supporting information

**S1 Fig. Negative binomial regression with complete cases.**
(PNG)

**S2 Fig. Negative binomial regression with multiple imputation on health literacy.** We applied multiple imputation on the total health literacy score. Then we repeated the negative binomial regression on the imputed dataset.
(PNG)

**S3 Fig. Negative binomial regression with averaging of two scores to impute an overall score for health literacy.** If participants missed only one health literacy score question, we used the average of the two non-missing scores to impute the missing health literacy score. Then we applied negative binomial regression with the complete cases.
(PNG)

**S4 Fig. Negative binomial regression with averaging of two scores and multiple imputation to impute an overall score for health literacy.** If participants missed only one health literacy score question, we used the average of the two non-missing scores to impute the missing health literacy score. Then we applied multiple imputation on the total health literacy score and repeated negative binomial regression on the imputed dataset.
(PNG)

**S5 Fig. Negative binomial regression with "prefer not to answer" and "don't know" as missing.** Multiple imputation was applied to health literacy as above, and the negative binomial regression method was applied on the imputed dataset.
(PNG)

**S6 Fig. Negative binomial regression with interaction terms.** Race/ethnicity interactions with sex/gender, age, and education were added to the model, significant interaction terms were shown in the forest plot.
(PNG)

## Acknowledgments

We wish to thank our participants who have joined *All of Us* and contributed to PPI, helped refine early materials, engaged in developing and evaluating PPI, and provided other ongoing feedback. We thank the countless other co-investigators and staff across all awardees and partners, without which *All of Us* would not have achieved our current goals.

*All of Us PPI Committee Members*: James McClain, Brian Ahmedani, Rob Cronin, Michael Manganiello, Kathy Mazor, Heather Sansbury, Alvaro Alonso, Sarra Hedden, Randy Bloom, Mick Couper, Scott Sutherland

We also wish to thank *All of Us* Research Program Director Josh Denny and our partners Verily, Vibrent, Scripps, and Leidos.

"Precision Medicine Initiative, PMI, All of Us, the *All of Us* logo, and The Future of Health Begins with You are service marks of the US Department of Health and Human Services."

## Author Contributions

**Writing – original draft:** Robert M. Cronin, Xiaoke Feng, Lina Sulieman, Brandy Mapes, Shawn Garbett, Ashley Able, Ryan Hale, Mick P. Couper, Heather Sansbury, Brian K. Ahmedani, Qingxia Chen.

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
