## [Decision Letter · Decision Letter 0]

2 Oct 2022

PONE-D-22-22898Missingness patterns in the baseline health surveys of the All of Us Research ProgramPLOS ONE

Dear Dr. Cronin,

Thank you for submitting your manuscript to PLOS ONE. After careful consideration, we feel that it has merit but does not (yet) meet PLOS ONE’s publication criteria as it currently stands. Therefore, we invite you to submit a revised version of the manuscript that addresses the points raised during the review process. I had invited two reviewers with vastly different backgrounds (substantive and with statistical expertise on missing data) to provide their insights. The comments from two independent reviewers are attached below, and I concur with both in the suggestion that a major revision is required to further consider the manuscript for publication in PLOS ONE. The manuscript title suggests a statistical approach while it doesn't seem to adopt one. The manuscript seems to be indecisive in terms of the angle taken (substantive or methodological), as also reflected by the comments of reviewer 1. In addition, reviewer 2 mentions substantive reasons and arguments for deeper and exhaustive elaborations on the meaning and interpretation of the patterns suggested in the manuscript. It is very important to support the suggested patterns with established techniques or, if not available, with novel statistical technique(s) for which evidence of validity and reliability are provided. For a paper to be considered in PLOS ONE - which has focus on methodological and statistical rigor -, the manuscript should provide a scope and message beyond (just) the current dataset or applications to it. Therefore, could you please revise the manuscript to clearly address the comments attached herein?

We look forward to receiving your revised manuscript.

Kind regards,

Ralph C. A. Rippe, Ph.D.

Academic Editor

PLOS ONE

Journal Requirements:

3. Thank you for stating the following in the Sources of financial support of your manuscript: 

"This work was funded by the National Institutes of Health (https://allofus.nih.gov/) 5U2COD023196 (RMC, XF, LS, BMM, SG, AS, RJ, MPC, QC) and 3OT2OD026550 (BKA). ( Additional support included the National Heart, Blood, and Lung Institute (https://www.nhlbi.nih.gov/) K23HL141447 (RMC)"

"This work was funded by the National Institutes of Health (https://allofus.nih.gov/) 5U2COD023196 (RMC, XF, LS, BMM, SG, AS, RJ, MPC, QC) and 3OT2OD026550 (BKA). (Additional support included the National Heart, Blood, and Lung Institute (https://www.nhlbi.nih.gov/) K23HL141447 (RMC). This team worked closely with the NIH on the study design approval and reviewed the manuscript for completeness."

Reviewers' comments:

Reviewer's Responses to Questions

**Comments to the Author**

1. Is the manuscript technically sound, and do the data support the conclusions?

Reviewer #1: Yes

Reviewer #2: Yes

2. Has the statistical analysis been performed appropriately and rigorously? 

Reviewer #1: I Don't Know

Reviewer #2: No

3. Have the authors made all data underlying the findings in their manuscript fully available?

Reviewer #1: Yes

Reviewer #2: Yes

4. Is the manuscript presented in an intelligible fashion and written in standard English?

Reviewer #1: Yes

Reviewer #2: Yes

5. Review Comments to the Author

Reviewer #1: General comment:

After reading the paper I thought: what is the goal, and what is the core message from the paper? If I understand correctly, the authors are trying to predict whether values on specific questions are missing, using other variables as predictors. Although that may be an interesting thing to look at, I think it would be way too limited to dedicate a complete paper to. Normally, checking whether missing data show a systematic pattern is part of the statistical analysis. I don’t see any statistical analysis here, apart from the prediction of the missing values.

Detailed comments:

1. P. 1, last line: “Various methods…”: This sounds kind of repetitive since earlier it was said that “There are multiple ways to handle missing data in general”. Maybe it should be emphasized more that here you are specifically talking about survey data.

2. P. 4, line 2: “Missing data can be examined by the participant”: at first it read like the examining of the missing data was done by the participant. Please, reformulate.

3. P. 5, line 4: although the advice used to be to impute 5 times, it was later established that 5 imputations is actually quite a small number (see, e.g., Graham, Olchowski, and Gilreath, 2007). I would advice increasing the number of imputations to 100.

4. P. 5, line 7: “Rubin’s rules” needs a reference.

5. I don’t understand the negative binomial regression, nor its purpose. If I understand correctly, a number of predictors is used to predict whether a value on an eligible question is missing or not. These predictors have missing data as well, and are multiply imputed as well? And why is prediction of missing data on the basis of background variables important?

6. I think the layout of the figures is occasionally sloppy. For example, in Figure 1, one of the two categories of age is labeled “>= 65” rather than “≥ 65”, in Figure 2 the variable names overlap with the graph, and “(a) Incidence Rate Ratio” overlaps with the numbers on the X axis.

References:

Graham, J.W., Olchowski, A.E., & Gilreath, T.D (2007). How many imputations are really needed? Some practical clarifications of multiple imputation theory. Prevention Science, 8, 206-213. doi: 10.1007/s11121-007-0070-9

Reviewer #2: I am very excited about this paper which will help a lot of future papers using All of Us data, so I am generally positive about this paper. My only issues are about more deep analysis of data on race/ethnicity. Here I explain:

1- Please also include data on missingness of race/ethnicity.

2- Based on your findings, please discuss if multiple imputation would reduce or increase bias due to missing data in this case.

3- Race/ethnicity interacts with sex/gender, age, and SES. For example, income is higher among high educated Whites than high educated Blacks, and Black men are treated worse by some institutions than Black women. This means, trust is missingness may not be just shaped by race but the interaction of race and other social constructs. Why this is important? Because missingness may vary not just by race and education, but race x education (due to reduced relevance of education in non-Whites). So, please kindly not only check the main effects of race, but also the interactions of race/ethnicity (at least for Blacks vs Whites) with education, age, and sex/gender on missingness.

Again, thank you for your service and exceptional work. Looking forward to cite your work in our future All of Us papers.

6. PLOS authors have the option to publish the peer review history of their article (what does this mean?). If published, this will include your full peer review and any attached files.

Reviewer #1: No

Reviewer #2: No

---

## [Author Response · Author response to Decision Letter 0]

24 Nov 2022

Editor Comments:

I had invited two reviewers with vastly different backgrounds (substantive and with statistical expertise on missing data) to provide their insights. The comments from two independent reviewers are attached below, and I concur with both in the suggestion that a major revision is required to further consider the manuscript for publication in PLOS ONE. The manuscript title suggests a statistical approach while it doesn't seem to adopt one. The manuscript seems to be indecisive in terms of the angle taken (substantive or methodological), as also reflected by the comments of reviewer 1. In addition, reviewer 2 mentions substantive reasons and arguments for deeper and exhaustive elaborations on the meaning and interpretation of the patterns suggested in the manuscript. It is very important to support the suggested patterns with established techniques or, if not available, with novel statistical technique(s) for which evidence of validity and reliability are provided. For a paper to be considered in PLOS ONE - which has focus on methodological and statistical rigor -, the manuscript should provide a scope and message beyond (just) the current dataset or applications to it. Therefore, could you please revise the manuscript to clearly address the comments attached herein?

>>>>>>>>

Response: 

We thank the Editor for their comments. We believe handling missingness is critical for statistical analyses of research questions. It is common for substantive researchers to ignore missing data, especially in large cohort studies where sample sizes are less of a concern. If variables are missing not at random (MNAR), complete case analysis or methods assuming missingness at random (MAR) could lead to biased conclusions. While the assumption on the missing mechanism is not testable, identifying factors associated with missingness is critical in lessening MNAR by controlling for those factors in the analysis. The findings presented in this manuscript can be useful to the researchers of the All of Us survey data and helps survey designers for other large cohorts to target strategies to reduce differential missingness. Similar analysis can be conducted for other large epidemiological cohorts like the Million Veterans Program or UK Biobank to help reduce potential biases and account for them. 

While this paper studies an important question that will impact the downstream decision process of statistical method development and implementation, the goal is not to develop novel statistical methods. We clarified the objective of this paper in the introduction:

The objective of this project was to use All of Us as a case study to demonstrate an approach to evaluate the missingness of surveys and identify characteristics associated with missingness in a large epidemiological cohort. In particular, we studied if the demographical variables that define the historically underrepresented groups in biomedical research, enrollment date, and health literacy were associated with an increased risk of missingness in the remaining survey questions of the baseline surveys. 

We also changed the title to “Importance of missingness in baseline variables: a case study of the All of Us Research Program.” 

 We appreciate the reviewer's comments and believe these comments have helped strengthen this manuscript significantly. We addressed the comments to better describe the angle for reviewer 1 comments and completed a deeper and exhaustive elaboration as recommended by reviewer 2. 

>>>>>>>>

>>>>>>>>

Response: We fixed the styles for PLOS ONE

>>>>>>>>

>>>>>>>>

Response: We included the correct grant numbers in the funding information section.

>>>>>>>>

3. Thank you for stating the following in the Sources of financial support of your manuscript: 

"This work was funded by the National Institutes of Health (https://allofus.nih.gov/) 5U2COD023196 (RMC, XF, LS, BMM, SG, AS, RJ, MPC, QC) and 3OT2OD026550 (BKA). ( Additional support included the National Heart, Blood, and Lung Institute (https://www.nhlbi.nih.gov/) K23HL141447 (RMC)"

"This work was funded by the National Institutes of Health (https://allofus.nih.gov/) 5U2COD023196 (RMC, XF, LS, BMM, SG, AS, RJ, MPC, QC) and 3OT2OD026550 (BKA). (Additional support included the National Heart, Blood, and Lung Institute (https://www.nhlbi.nih.gov/) K23HL141447 (RMC). This team worked closely with the NIH on the study design approval and reviewed the manuscript for completeness."

>>>>>>>>

Response: The above funding statement is correct; we removed this information from the acknowledgments.

>>>>>>>>

>>>>>>>>

Response: We updated the data availability statement in the submission:

Data is owned by a third party, the All of Us Research Program. The data underlying this article were provided by the All of Us Research Program by permission that can be sought by scientists and the public alike. The Researcher Workbench is a cloud-based platform where registered researchers can access Registered and Controlled Tier data, including the data presented here. Researchers/citizen scientists must verify their identity and complete the All of Us Research Program data access process to access the Researcher Workbench and Registered Tier data. Once this process is completed, the data will be made available to all persons. More information on data access can be found in the All of Us Research Hub (https://www.researchallofus.org/) as is the option to register for access. The authors did not have any special access privileges to this data that other researchers would not have.

>>>>>>>>

>>>>>>>>

Response: We added the full ethics statement to the method section of the manuscript.

>>>>>>>>

Reviewer #1: General comment:

After reading the paper I thought: what is the goal, and what is the core message from the paper? If I understand correctly, the authors are trying to predict whether values on specific questions are missing, using other variables as predictors. Although that may be an interesting thing to look at, I think it would be way too limited to dedicate a complete paper to. Normally, checking whether missing data show a systematic pattern is part of the statistical analysis. I don’t see any statistical analysis here, apart from the prediction of the missing values.

>>>>>>>>

Response: We apologize for any confusion. The objective of this article is to use All of Us as a case study to demonstrate an approach to evaluate the missingness of surveys and identify key variables, in particular, the variables defining underrepresented groups in biomedical research, that are associated with missingness in a large epidemiological cohort. Handling missingness is critical for additional statistical analyses of research questions. It is common for substantive researchers to ignore missing data, especially in large cohort studies where small sample sizes are less of a concern. If variables are missing not at random (MNAR), complete case analysis or methods assuming missing completely at random (MCAR) could lead to biased conclusions. While the assumption on the missing mechanism is not testable, identifying factors associated with missingness is critical in lessening MNAR by controlling for those factors in the analysis. We agree that checking missing data mechanisms, especially the individual question related to the specific research question is, and has to be, part of its own statistical analysis. However, pre-identifying common variables explaining the missingness of the other variables will reduce the analysis burden in the substantive analysis, especially when the size of potential variables to evaluate is large (a total of 126 questions from the three baseline modules) and branching logic is involved as in our study. As we clarified in our response to the Editor, while this paper studies an important question that will impact the downstream decision process of statistical method development and implementation, the goal is not to develop novel statistical methods. We added the following to the introduction:

The objective of this project was to use All of Us as a case study to demonstrate an approach to evaluate the missingness of surveys and identify characteristics associated with missingness in a large epidemiological cohort. In particular, we studied if the demographical variables that define the historically underrepresented groups in biomedical research, enrollment date, and health literacy were associated with an increased risk of missingness in the remaining survey questions of the baseline surveys. 

We also changed the title to “Importance of missingness in baseline variables: a case study of the All of Us Research Program.”

For All of Us, the three baseline modules studied in this article were required for all participants and contained the information that could be commonly used in the All of Us research program. In this paper, we described the completeness of these survey data and identified some key variables associated with the overall missingness of the rest of the variables from the three baseline survey modules. The fact that those often not included in biomedical research have higher rates of missingness is a potential red flag for studies such as All of Us, which make special efforts to include such populations. If these populations fail to answer other questions or drop out altogether, any complete case analysis will mean that these losses undermine the efforts to include such groups. Another key message of our paper is identifying "leading indicators" of missingness (i.e., the predictors) that could help survey designers to target strategies to reduce differential missingness. 

The findings presented in this manuscript can be useful to the researchers of the All of Us survey data and helps survey designers for other large cohorts to target strategies to reduce differential missingness. Similar analysis can be conducted for other large epidemiological cohorts like the Million Veterans Program or UK Biobank to help reduce potential biases and account for them. The fact that we found a low missing percentage in the three baseline survey modules of the All of Us cohort offers some reassurance to substantive research but also points to the importance of adjusting for the key variables associated with overall missingness on substantive analyses.

We added the following to the discussion: 

Discussion:

For All of Us, the three baseline modules studied in this article were required for all participants and contained the information that could be commonly used in the All of Us research program. In this manuscript, we described the completeness of these survey data and identified some key baseline variables associated with the overall missingness of the rest of the variables from the three baseline survey modules. The fact that those often not included in biomedical research have higher rates of missingness is a potential red flag for studies, such as All of Us, that make special efforts to include such populations. The missingness of these background variables of race, sex and gender identity, education, time of enrollment, and geography, could contribute to non-random missingness in surveys. If these populations fail to answer other questions or drop out altogether, any complete case analysis will mean that these losses undermine the efforts to include such groups. 

[…]

The analyses presented in this manuscript can be used by researchers for the All of Us survey data and survey data from other large epidemiological cohorts like the Million Veterans Program or UK Biobank to help reduce potential biases and account for them. Another key message is that identifying “leading indicators” of missingness (i.e., the predictors) could help survey designers to target strategies to reduce differential missingness. The fact that we found a low missing percentage in the three baseline survey modules of the All of Us cohort offers some reassurance to substantive research but also points to the importance of adjusting for the critical variables associated with overall missingness on substantive analyses. Researchers must be cautious when using complete case analysis or assuming missingness at random in All of Us or other large epidemiological cohorts (e.g., UK biobank, Million Veterans Program). 

>>>>>>>>

Detailed comments:

1. P. 1, last line: “Various methods…”: This sounds kind of repetitive since earlier it was said that “There are multiple ways to handle missing data in general”. Maybe it should be emphasized more that here you are specifically talking about survey data.

>>>>>>>>

Response: We appreciate the comment and modified the introduction as follows: 

“There are multiple ways to handle missing survey data in general.”

>>>>>>>>

2. P. 4, line 2: “Missing data can be examined by the participant”: at first it read like the examining of the missing data was done by the participant. Please, reformulate.

>>>>>>>>

Response: We appreciate the reviewer pointing out this potential confusion. We modified to the following: 

“Missing data can be examined at the level of the participant.”

>>>>>>>>

3. P. 5, line 4: although the advice used to be to impute 5 times, it was later established that 5 imputations are actually quite a small number (see, e.g., Graham, Olchowski, and Gilreath, 2007). I would advise increasing the number of imputations to 100.

>>>>>>>>

Response: We appreciate the reviewer’s comment, reviewed the reference mentioned and Rubin[1], and included both pieces of information below. We imputed the health literacy score, which had about 6% (0.06) missing.

Graham et al. (2007) conclusions state: “With these assumptions, we recommend that one should use m = 20, 20, 40, 100, and >100 for true γ = 0.10, 0.30, 0.50, 0.70, and 0.90, respectively,” where the fraction of missing information (γ). Although γ is the same as the amount of missing data in the simplest case, it is typically rather less than the amount of missing data, per se, in more complicated situations (Rubin[1]). 

Also, in Graham et al. (2007):

“Rubin[1] shows that the efficiency of an estimate based on m imputations is approximately 

(1+γ/m)^[2], where γ is the fraction of missing information for the quantity being estimated.... gains rapidly diminish after the first few imputations. ... In most situations, there is simply little advantage to producing and analyzing more than a few imputed datasets (pp. 548–549).”

The recommendation of 100*missing percentage imputation comes from this formula allowing a 1% efficiency loss. With 6% missing in the data for health literacy, which we imputed, γ is less than 0.06. Therefore, 6 imputations lead to <1% efficiency loss, 10 imputations lead to <0.6% efficiency loss and 100 imputations lead to <0.06% efficiency loss. 

Considering this above and the computation burden for the large data size, the other factor that Graham et al. (2007) pointed out, we increased the number of imputations to 6, which gives us a <1% efficiency loss.

We updated the methods and results in the manuscript and included Graham et al. (2007) and Rubin in the reference:

Methods:

The health literacy scale had about 6% missingness. Therefore, we used six imputations to allow for a <1% efficiency loss[1, 3].

>>>>>>>>

4. P. 5, line 7: “Rubin’s rules” needs a reference.

>>>>>>>>

Response: We added the following reference[1]:

1. Rubin DB. Multiple imputation for nonresponse in surveys. Vol. 81: John Wiley & Sons; 2004.

>>>>>>>>

5. I don’t understand the negative binomial regression, nor its purpose. 

>>>>>>>>

Response: We used negative binomial regression because the dependent variable is the number of questions (excluding the questions included as the independent variables) with a missing value out of the total number of eligible questions based on branching logic (each participant could have a different number of eligible questions). Instead of Poisson regression, negative binomial regression was used to allow for over-dispersion.

>>>>>>>>

5. If I understand correctly, a number of predictors is used to predict whether a value on an eligible question is missing or not. 

>>>>>>>>

Response: Predictors were used to study their associations with the level of missingness for an eligible participant 

We updated the methods as follows to improve clarity:

We performed a negative binomial regression to evaluate the impact of participant characteristics on the percentage of missingness of a participant based on the number of eligible questions for a participant allowing for overdispersion

>>>>>>>>

5. These predictors have missing data as well, and are multiply imputed as well? 

>>>>>>>>

Response: Multiple imputation was only used for the health literacy scale. One of the key findings of this paper is that those who skip key demographic questions are more likely to skip other substantive variables in the surveys (see Figure 1). Imputing these predictor variables would mask this relationship. These variables could also be a leading indicator of later missingness in the surveys. We also updated the regression to evaluate missingness from the remaining questions and not include the predictors in the outcomes.

>>>>>>>>

5. And why is prediction of missing data on the basis of background variables important?

>>>>>>>>

Response: 

The missingness of background variables of race, sex and gender identity, education, time of enrollment, and geography, could contribute to non-random missingness in surveys. Issues of missingness based on these background variables have affected surveys from other national programs such as NHANES and the UK biobank[4-7]. Of the many papers (over 75) that have used the UK biobank, we could only find two that explicitly discussed missing data[6, 7]. Our manuscript helps highlight the importance of evaluating this potential issue for All of Us and other data sets with survey data. Using complete case analysis or assuming missingness at random, without evaluating background variables, could bias the research conclusions described above in response to the general comment. Researchers must be cautious when using complete case analysis or assuming missingness at random in All of Us or other large epidemiological cohorts (e.g., UK biobank, Million Veterans Program). 

We added the following to the manuscript, as mentioned above to the response to the general comment from reviewer #1:

For All of Us, the three baseline modules studied in this article were required for all participants and contained the information that could be commonly used in the All of Us research program. In this manuscript, we described the completeness of these survey data and identified some key baseline variables associated with the overall missingness of the rest of the variables from the three baseline survey modules. The fact that those often not included in biomedical research have higher rates of missingness is a potential red flag for studies, such as All of Us, that make special efforts to include such populations. The missingness of these background variables of race, sex and gender identity, education, time of enrollment, and geography, could contribute to non-random missingness in surveys. If these populations fail to answer other questions or drop out altogether, any complete case analysis will mean that these losses undermine the efforts to include such groups. 

[…]

The analyses presented in this manuscript can be used by researchers for the All of Us survey data and survey data from other large epidemiological cohorts like the Million Veterans Program or UK Biobank to help reduce potential biases and account for them. Another key message is that identifying “leading indicators” of missingness (i.e., the predictors) could help survey designers to target strategies to reduce differential missingness. The fact that we found a low missing percentage in the three baseline survey modules of the All of Us cohort offers some reassurance to substantive research but also points to the importance of adjusting for the critical variables associated with overall missingness on substantive analyses. Researchers must be cautious when using complete case analysis or assuming missingness at random in All of Us or other large epidemiological cohorts (e.g., UK biobank, Million Veterans Program). 

>>>>>>>>

6. I think the layout of the figures is occasionally sloppy. For example, in Figure 1, one of the two categories of age is labeled “>= 65” rather than “≥ 65”, in Figure 2 the variable names overlap with the graph, and “(a) Incidence Rate Ratio” overlaps with the numbers on the X axis.

>>>>>>>>

Response: We appreciate the reviewer’s comments and have improved the layout of the figures.

>>>>>>>>

References:

Graham, J.W., Olchowski, A.E., & Gilreath, T.D (2007). How many imputations are really needed? Some practical clarifications of multiple imputation theory. Prevention Science, 8, 206-213. doi: 10.1007/s11121-007-0070-9

Reviewer #2: I am very excited about this paper which will help a lot of future papers using All of Us data, so I am generally positive about this paper. My only issues are about more deep analysis of data on race/ethnicity. Here I explain:

1- Please also include data on missingness of race/ethnicity.

>>>>>>>>

Response: Missingness of the question for race/ethnicity is included in Figure 1a. We considered missing data for this question to be a skip or a “prefer not to answer” response. 

Overall we saw 5741 (1.7%) with missing data for race/ethnicity. We also added this to the results: 

For example, 5741 participants (1.7%) who did not answer a race/ethnicity skipped 13% of the remaining questions of the baseline surveys, While the average skip rate was about 5% of the questions. 

>>>>>>>>

2- Based on your findings, please discuss if multiple imputation would reduce or increase bias due to missing data in this case.

>>>>>>>>

Response: When the missing at random assumption is valid, multiple imputation will reduce the bias and lead to a more valid inference. 

>>>>>>>>

3- Race/ethnicity interacts with sex/gender, age, and SES. For example, income is higher among high educated Whites than high educated Blacks, and Black men are treated worse by some institutions than Black women. This means, trust is missingness may not be just shaped by race but the interaction of race and other social constructs. Why this is important? Because missingness may vary not just by race and education, but race x education (due to reduced relevance of education in non-Whites). So, please kindly not only check the main effects of race, but also the interactions of race/ethnicity (at least for Blacks vs. Whites) with education, age, and sex/gender on missingness.

>>>>>>>>

Response: We greatly appreciate this suggestion by the reviewer. We added the interaction terms to our primary analysis and included the following in the methods:

As race/ethnicity was associated with an increased risk of missingness, we also investigated if the effect was moderated by education, age, and sex/gender by including the two-way interaction terms in the model.

We added an appendix figure showing only the effects of the interaction terms that were significant and the following to the results: 

Some race/ethnicity interactions with sex/gender, age, and education were significant and demonstrated differential race/ethnicity effects moderated other baseline variables (see the supplemental document, Figs. S6). However, the most significant IRRs were among the groups with missingness in the baseline variables.

>>>>>>>>

Again, thank you for your service and exceptional work. Looking forward to cite your work in our future All of Us papers.

>>>>>>>>

Response: Thank you very much for your review! 

>>>>>>>>

REFERENCES

1. Rubin DB. Multiple imputation for nonresponse in surveys: John Wiley & Sons; 2004.

2. Löwe B, Kroenke K, Herzog W, Gräfe K. Measuring depression outcome with a brief self-report instrument: sensitivity to change of the Patient Health Questionnaire (PHQ-9). Journal of affective disorders. 2004;81(1):61-6. doi: https://doi.org/10.1016/S0165-0327(03)00198-8.

3. Graham JW, Olchowski AE, Gilreath TD. How many imputations are really needed? Some practical clarifications of multiple imputation theory. Prevention science. 2007;8(3):206-13.

4. Hartwell ML, Khojasteh J, Wetherill MS, Croff JM, Wheeler D. Using Structural Equation Modeling to Examine the Influence of Social, Behavioral, and Nutritional Variables on Health Outcomes Based on NHANES Data: Addressing Complex Design, Nonnormally Distributed Variables, and Missing Information. Current Developments in Nutrition. 2019;3(5). doi: 10.1093/cdn/nzz010.

5. Pridham G, Rockwood K, Rutenberg A. Strategies for handling missing data that improve Frailty Index estimation and predictive power: lessons from the NHANES dataset. GeroScience. 2022;44(2):897-923. doi: 10.1007/s11357-021-00489-w.

6. Lv X, Li Y, Li R, Guan X, Li L, Li J, et al. Relationships of sleep traits with prostate cancer risk: A prospective study of 213,999 UK Biobank participants. The Prostate. 2022;82(9):984-92. doi: https://doi.org/10.1002/pros.24345.

7. Foster HME, Celis-Morales CA, Nicholl BI, Petermann-Rocha F, Pell JP, Gill JMR, et al. The effect of socioeconomic deprivation on the association between an extended measurement of unhealthy lifestyle factors and health outcomes: a prospective analysis of the UK Biobank cohort. The Lancet Public Health. 2018;3(12):e576-e85. doi: 10.1016/S2468-2667(18)30200-7.

Editor Comments:

I had invited two reviewers with vastly different backgrounds (substantive and with statistical expertise on missing data) to provide their insights. The comments from two independent reviewers are attached below, and I concur with both in the suggestion that a major revision is required to further consider the manuscript for publication in PLOS ONE. The manuscript title suggests a statistical approach while it doesn't seem to adopt one. The manuscript seems to be indecisive in terms of the angle taken (substantive or methodological), as also reflected by the comments of reviewer 1. In addition, reviewer 2 mentions substantive reasons and arguments for deeper and exhaustive elaborations on the meaning and interpretation of the patterns suggested in the manuscript. It is very important to support the suggested patterns with established techniques or, if not available, with novel statistical technique(s) for which evidence of validity and reliability are provided. For a paper to be considered in PLOS ONE - which has focus on methodological and statistical rigor -, the manuscript should provide a scope and message beyond (just) the current dataset or applications to it. Therefore, could you please revise the manuscript to clearly address the comments attached herein?

>>>>>>>>

Response: 

We thank the Editor for their comments. We believe handling missingness is critical for statistical analyses of research questions. It is common for substantive researchers to ignore missing data, especially in large cohort studies where sample sizes are less of a concern. If variables are missing not at random (MNAR), complete case analysis or methods assuming missingness at random (MAR) could lead to biased conclusions. While the assumption on the missing mechanism is not testable, identifying factors associated with missingness is critical in lessening MNAR by controlling for those factors in the analysis. The findings presented in this manuscript can be useful to the researchers of the All of Us survey data and helps survey designers for other large cohorts to target strategies to reduce differential missingness. Similar analysis can be conducted for other large epidemiological cohorts like the Million Veterans Program or UK Biobank to help reduce potential biases and account for them. 

While this paper studies an important question that will impact the downstream decision process of statistical method development and implementation, the goal is not to develop novel statistical methods. We clarified the objective of this paper in the introduction:

The objective of this project was to use All of Us as a case study to demonstrate an approach to evaluate the missingness of surveys and identify characteristics associated with missingness in a large epidemiological cohort. In particular, we studied if the demographical variables that define the historically underrepresented groups in biomedical research, enrollment date, and health literacy were associated with an increased risk of missingness in the remaining survey questions of the baseline surveys. 

We also changed the title to “Importance of missingness in baseline variables: a case study of the All of Us Research Program.” 

 We appreciate the reviewer's comments and believe these comments have helped strengthen this manuscript significantly. We addressed the comments to better describe the angle for reviewer 1 comments and completed a deeper and exhaustive elaboration as recommended by reviewer 2. 

>>>>>>>>

>>>>>>>>

Response: We fixed the styles for PLOS ONE

>>>>>>>>

>>>>>>>>

Response: We included the correct grant numbers in the funding information section.

>>>>>>>>

3. Thank you for stating the following in the Sources of financial support of your manuscript: 

"This work was funded by the National Institutes of Health (https://allofus.nih.gov/) 5U2COD023196 (RMC, XF, LS, BMM, SG, AS, RJ, MPC, QC) and 3OT2OD026550 (BKA). ( Additional support included the National Heart, Blood, and Lung Institute (https://www.nhlbi.nih.gov/) K23HL141447 (RMC)"

"This work was funded by the National Institutes of Health (https://allofus.nih.gov/) 5U2COD023196 (RMC, XF, LS, BMM, SG, AS, RJ, MPC, QC) and 3OT2OD026550 (BKA). (Additional support included the National Heart, Blood, and Lung Institute (https://www.nhlbi.nih.gov/) K23HL141447 (RMC). This team worked closely with the NIH on the study design approval and reviewed the manuscript for completeness."

>>>>>>>>

Response: The above funding statement is correct; we removed this information from the acknowledgments.

>>>>>>>>

>>>>>>>>

Response: We updated the data availability statement in the submission:

Data is owned by a third party, the All of Us Research Program. The data underlying this article were provided by the All of Us Research Program by permission that can be sought by scientists and the public alike. The Researcher Workbench is a cloud-based platform where registered researchers can access Registered and Controlled Tier data, including the data presented here. Researchers/citizen scientists must verify their identity and complete the All of Us Research Program data access process to access the Researcher Workbench and Registered Tier data. Once this process is completed, the data will be made available to all persons. More information on data access can be found in the All of Us Research Hub (https://www.researchallofus.org/) as is the option to register for access. The authors did not have any special access privileges to this data that other researchers would not have.

>>>>>>>>

>>>>>>>>

Response: We added the full ethics statement to the method section of the manuscript.

>>>>>>>>

Reviewer #1: General comment:

After reading the paper I thought: what is the goal, and what is the core message from the paper? If I understand correctly, the authors are trying to predict whether values on specific questions are missing, using other variables as predictors. Although that may be an interesting thing to look at, I think it would be way too limited to dedicate a complete paper to. Normally, checking whether missing data show a systematic pattern is part of the statistical analysis. I don’t see any statistical analysis here, apart from the prediction of the missing values.

>>>>>>>>

Response: We apologize for any confusion. The objective of this article is to use All of Us as a case study to demonstrate an approach to evaluate the missingness of surveys and identify key variables, in particular, the variables defining underrepresented groups in biomedical research, that are associated with missingness in a large epidemiological cohort. Handling missingness is critical for additional statistical analyses of research questions. It is common for substantive researchers to ignore missing data, especially in large cohort studies where small sample sizes are less of a concern. If variables are missing not at random (MNAR), complete case analysis or methods assuming missing completely at random (MCAR) could lead to biased conclusions. While the assumption on the missing mechanism is not testable, identifying factors associated with missingness is critical in lessening MNAR by controlling for those factors in the analysis. We agree that checking missing data mechanisms, especially the individual question related to the specific research question is, and has to be, part of its own statistical analysis. However, pre-identifying common variables explaining the missingness of the other variables will reduce the analysis burden in the substantive analysis, especially when the size of potential variables to evaluate is large (a total of 126 questions from the three baseline modules) and branching logic is involved as in our study. As we clarified in our response to the Editor, while this paper studies an important question that will impact the downstream decision process of statistical method development and implementation, the goal is not to develop novel statistical methods. We added the following to the introduction:

The objective of this project was to use All of Us as a case study to demonstrate an approach to evaluate the missingness of surveys and identify characteristics associated with missingness in a large epidemiological cohort. In particular, we studied if the demographical variables that define the historically underrepresented groups in biomedical research, enrollment date, and health literacy were associated with an increased risk of missingness in the remaining survey questions of the baseline surveys. 

We also changed the title to “Importance of missingness in baseline variables: a case study of the All of Us Research Program.”

For All of Us, the three baseline modules studied in this article were required for all participants and contained the information that could be commonly used in the All of Us research program. In this paper, we described the completeness of these survey data and identified some key variables associated with the overall missingness of the rest of the variables from the three baseline survey modules. The fact that those often not included in biomedical research have higher rates of missingness is a potential red flag for studies such as All of Us, which make special efforts to include such populations. If these populations fail to answer other questions or drop out altogether, any complete case analysis will mean that these losses undermine the efforts to include such groups. Another key message of our paper is identifying "leading indicators" of missingness (i.e., the predictors) that could help survey designers to target strategies to reduce differential missingness. 

The findings presented in this manuscript can be useful to the researchers of the All of Us survey data and helps survey designers for other large cohorts to target strategies to reduce differential missingness. Similar analysis can be conducted for other large epidemiological cohorts like the Million Veterans Program or UK Biobank to help reduce potential biases and account for them. The fact that we found a low missing percentage in the three baseline survey modules of the All of Us cohort offers some reassurance to substantive research but also points to the importance of adjusting for the key variables associated with overall missingness on substantive analyses.

We added the following to the discussion: 

Discussion:

For All of Us, the three baseline modules studied in this article were required for all participants and contained the information that could be commonly used in the All of Us research program. In this manuscript, we described the completeness of these survey data and identified some key baseline variables associated with the overall missingness of the rest of the variables from the three baseline survey modules. The fact that those often not included in biomedical research have higher rates of missingness is a potential red flag for studies, such as All of Us, that make special efforts to include such populations. The missingness of these background variables of race, sex and gender identity, education, time of enrollment, and geography, could contribute to non-random missingness in surveys. If these populations fail to answer other questions or drop out altogether, any complete case analysis will mean that these losses undermine the efforts to include such groups. 

[…]

The analyses presented in this manuscript can be used by researchers for the All of Us survey data and survey data from other large epidemiological cohorts like the Million Veterans Program or UK Biobank to help reduce potential biases and account for them. Another key message is that identifying “leading indicators” of missingness (i.e., the predictors) could help survey designers to target strategies to reduce differential missingness. The fact that we found a low missing percentage in the three baseline survey modules of the All of Us cohort offers some reassurance to substantive research but also points to the importance of adjusting for the critical variables associated with overall missingness on substantive analyses. Researchers must be cautious when using complete case analysis or assuming missingness at random in All of Us or other large epidemiological cohorts (e.g., UK biobank, Million Veterans Program). 

>>>>>>>>

Detailed comments:

1. P. 1, last line: “Various methods…”: This sounds kind of repetitive since earlier it was said that “There are multiple ways to handle missing data in general”. Maybe it should be emphasized more that here you are specifically talking about survey data.

>>>>>>>>

Response: We appreciate the comment and modified the introduction as follows: 

“There are multiple ways to handle missing survey data in general.”

>>>>>>>>

2. P. 4, line 2: “Missing data can be examined by the participant”: at first it read like the examining of the missing data was done by the participant. Please, reformulate.

>>>>>>>>

Response: We appreciate the reviewer pointing out this potential confusion. We modified to the following: 

“Missing data can be examined at the level of the participant.”

>>>>>>>>

3. P. 5, line 4: although the advice used to be to impute 5 times, it was later established that 5 imputations are actually quite a small number (see, e.g., Graham, Olchowski, and Gilreath, 2007). I would advise increasing the number of imputations to 100.

>>>>>>>>

Response: We appreciate the reviewer’s comment, reviewed the reference mentioned and Rubin[1], and included both pieces of information below. We imputed the health literacy score, which had about 6% (0.06) missing.

Graham et al. (2007) conclusions state: “With these assumptions, we recommend that one should use m = 20, 20, 40, 100, and >100 for true γ = 0.10, 0.30, 0.50, 0.70, and 0.90, respectively,” where the fraction of missing information (γ). Although γ is the same as the amount of missing data in the simplest case, it is typically rather less than the amount of missing data, per se, in more complicated situations (Rubin[1]). 

Also, in Graham et al. (2007):

“Rubin[1] shows that the efficiency of an estimate based on m imputations is approximately 

(1+γ/m)^[2], where γ is the fraction of missing information for the quantity being estimated.... gains rapidly diminish after the first few imputations. ... In most situations, there is simply little advantage to producing and analyzing more than a few imputed datasets (pp. 548–549).”

The recommendation of 100*missing percentage imputation comes from this formula allowing a 1% efficiency loss. With 6% missing in the data for health literacy, which we imputed, γ is less than 0.06. Therefore, 6 imputations lead to <1% efficiency loss, 10 imputations lead to <0.6% efficiency loss and 100 imputations lead to <0.06% efficiency loss. 

Considering this above and the computation burden for the large data size, the other factor that Graham et al. (2007) pointed out, we increased the number of imputations to 6, which gives us a <1% efficiency loss.

We updated the methods and results in the manuscript and included Graham et al. (2007) and Rubin in the reference:

Methods:

The health literacy scale had about 6% missingness. Therefore, we used six imputations to allow for a <1% efficiency loss[1, 3].

>>>>>>>>

4. P. 5, line 7: “Rubin’s rules” needs a reference.

>>>>>>>>

Response: We added the following reference[1]:

1. Rubin DB. Multiple imputation for nonresponse in surveys. Vol. 81: John Wiley & Sons; 2004.

>>>>>>>>

5. I don’t understand the negative binomial regression, nor its purpose. 

>>>>>>>>

Response: We used negative binomial regression because the dependent variable is the number of questions (excluding the questions included as the independent variables) with a missing value out of the total number of eligible questions based on branching logic (each participant could have a different number of eligible questions). Instead of Poisson regression, negative binomial regression was used to allow for over-dispersion.

>>>>>>>>

5. If I understand correctly, a number of predictors is used to predict whether a value on an eligible question is missing or not. 

>>>>>>>>

Response: Predictors were used to study their associations with the level of missingness for an eligible participant 

We updated the methods as follows to improve clarity:

We performed a negative binomial regression to evaluate the impact of participant characteristics on the percentage of missingness of a participant based on the number of eligible questions for a participant allowing for overdispersion

>>>>>>>>

5. These predictors have missing data as well, and are multiply imputed as well? 

>>>>>>>>

Response: Multiple imputation was only used for the health literacy scale. One of the key findings of this paper is that those who skip key demographic questions are more likely to skip other substantive variables in the surveys (see Figure 1). Imputing these predictor variables would mask this relationship. These variables could also be a leading indicator of later missingness in the surveys. We also updated the regression to evaluate missingness from the remaining questions and not include the predictors in the outcomes.

>>>>>>>>

5. And why is prediction of missing data on the basis of background variables important?

>>>>>>>>

Response: 

The missingness of background variables of race, sex and gender identity, education, time of enrollment, and geography, could contribute to non-random missingness in surveys. Issues of missingness based on these background variables have affected surveys from other national programs such as NHANES and the UK biobank[4-7]. Of the many papers (over 75) that have used the UK biobank, we could only find two that explicitly discussed missing data[6, 7]. Our manuscript helps highlight the importance of evaluating this potential issue for All of Us and other data sets with survey data. Using complete case analysis or assuming missingness at random, without evaluating background variables, could bias the research conclusions described above in response to the general comment. Researchers must be cautious when using complete case analysis or assuming missingness at random in All of Us or other large epidemiological cohorts (e.g., UK biobank, Million Veterans Program). 

We added the following to the manuscript, as mentioned above to the response to the general comment from reviewer #1:

For All of Us, the three baseline modules studied in this article were required for all participants and contained the information that could be commonly used in the All of Us research program. In this manuscript, we described the completeness of these survey data and identified some key baseline variables associated with the overall missingness of the rest of the variables from the three baseline survey modules. The fact that those often not included in biomedical research have higher rates of missingness is a potential red flag for studies, such as All of Us, that make special efforts to include such populations. The missingness of these background variables of race, sex and gender identity, education, time of enrollment, and geography, could contribute to non-random missingness in surveys. If these populations fail to answer other questions or drop out altogether, any complete case analysis will mean that these losses undermine the efforts to include such groups. 

[…]

The analyses presented in this manuscript can be used by researchers for the All of Us survey data and survey data from other large epidemiological cohorts like the Million Veterans Program or UK Biobank to help reduce potential biases and account for them. Another key message is that identifying “leading indicators” of missingness (i.e., the predictors) could help survey designers to target strategies to reduce differential missingness. The fact that we found a low missing percentage in the three baseline survey modules of the All of Us cohort offers some reassurance to substantive research but also points to the importance of adjusting for the critical variables associated with overall missingness on substantive analyses. Researchers must be cautious when using complete case analysis or assuming missingness at random in All of Us or other large epidemiological cohorts (e.g., UK biobank, Million Veterans Program). 

>>>>>>>>

6. I think the layout of the figures is occasionally sloppy. For example, in Figure 1, one of the two categories of age is labeled “>= 65” rather than “≥ 65”, in Figure 2 the variable names overlap with the graph, and “(a) Incidence Rate Ratio” overlaps with the numbers on the X axis.

>>>>>>>>

Response: We appreciate the reviewer’s comments and have improved the layout of the figures.

>>>>>>>>

References:

Graham, J.W., Olchowski, A.E., & Gilreath, T.D (2007). How many imputations are really needed? Some practical clarifications of multiple imputation theory. Prevention Science, 8, 206-213. doi: 10.1007/s11121-007-0070-9

Reviewer #2: I am very excited about this paper which will help a lot of future papers using All of Us data, so I am generally positive about this paper. My only issues are about more deep analysis of data on race/ethnicity. Here I explain:

1- Please also include data on missingness of race/ethnicity.

>>>>>>>>

Response: Missingness of the question for race/ethnicity is included in Figure 1a. We considered missing data for this question to be a skip or a “prefer not to answer” response. 

Overall we saw 5741 (1.7%) with missing data for race/ethnicity. We also added this to the results: 

For example, 5741 participants (1.7%) who did not answer a race/ethnicity skipped 13% of the remaining questions of the baseline surveys, While the average skip rate was about 5% of the questions. 

>>>>>>>>

2- Based on your findings, please discuss if multiple imputation would reduce or increase bias due to missing data in this case.

>>>>>>>>

Response: When the missing at random assumption is valid, multiple imputation will reduce the bias and lead to a more valid inference. 

>>>>>>>>

3- Race/ethnicity interacts with sex/gender, age, and SES. For example, income is higher among high educated Whites than high educated Blacks, and Black men are treated worse by some institutions than Black women. This means, trust is missingness may not be just shaped by race but the interaction of race and other social constructs. Why this is important? Because missingness may vary not just by race and education, but race x education (due to reduced relevance of education in non-Whites). So, please kindly not only check the main effects of race, but also the interactions of race/ethnicity (at least for Blacks vs. Whites) with education, age, and sex/gender on missingness.

>>>>>>>>

Response: We greatly appreciate this suggestion by the reviewer. We added the interaction terms to our primary analysis and included the following in the methods:

As race/ethnicity was associated with an increased risk of missingness, we also investigated if the effect was moderated by education, age, and sex/gender by including the two-way interaction terms in the model.

We added an appendix figure showing only the effects of the interaction terms that were significant and the following to the results: 

Some race/ethnicity interactions with sex/gender, age, and education were significant and demonstrated differential race/ethnicity effects moderated other baseline variables (see the supplemental document, Figs. S6). However, the most significant IRRs were among the groups with missingness in the baseline variables.

>>>>>>>>

Again, thank you for your service and exceptional work. Looking forward to cite your work in our future All of Us papers.

>>>>>>>>

Response: Thank you very much for your review! 

>>>>>>>>

REFERENCES

1. Rubin DB. Multiple imputation for nonresponse in surveys: John Wiley & Sons; 2004.

2. Löwe B, Kroenke K, Herzog W, Gräfe K. Measuring depression outcome with a brief self-report instrument: sensitivity to change of the Patient Health Questionnaire (PHQ-9). Journal of affective disorders. 2004;81(1):61-6. doi: https://doi.org/10.1016/S0165-0327(03)00198-8.

3. Graham JW, Olchowski AE, Gilreath TD. How many imputations are really needed? Some practical clarifications of multiple imputation theory. Prevention science. 2007;8(3):206-13.

4. Hartwell ML, Khojasteh J, Wetherill MS, Croff JM, Wheeler D. Using Structural Equation Modeling to Examine the Influence of Social, Behavioral, and Nutritional Variables on Health Outcomes Based on NHANES Data: Addressing Complex Design, Nonnormally Distributed Variables, and Missing Information. Current Developments in Nutrition. 2019;3(5). doi: 10.1093/cdn/nzz010.

5. Pridham G, Rockwood K, Rutenberg A. Strategies for handling missing data that improve Frailty Index estimation and predictive power: lessons from the NHANES dataset. GeroScience. 2022;44(2):897-923. doi: 10.1007/s11357-021-00489-w.

6. Lv X, Li Y, Li R, Guan X, Li L, Li J, et al. Relationships of sleep traits with prostate cancer risk: A prospective study of 213,999 UK Biobank participants. The Prostate. 2022;82(9):984-92. doi: https://doi.org/10.1002/pros.24345.

7. Foster HME, Celis-Morales CA, Nicholl BI, Petermann-Rocha F, Pell JP, Gill JMR, et al. The effect of socioeconomic deprivation on the association between an extended measurement of unhealthy lifestyle factors and health outcomes: a prospective analysis of the UK Biobank cohort. The Lancet Public Health. 2018;3(12):e576-e85. doi: 10.1016/S2468-2667(18)30200-7.

---

## [Decision Letter · Decision Letter 1]

3 May 2023

Importance of missingness in baseline variables: a case study of the All of Us Research Program

PONE-D-22-22898R1

Dear Dr. Cronin,

We’re pleased to inform you that your manuscript has been judged scientifically suitable for publication and will be formally accepted for publication once it meets all outstanding technical requirements.

Kind regards,

Bijan Najafi

Academic Editor

PLOS ONE

Additional Editor Comments (optional):

Thank you for your diligent work in revising the manuscript following the initial rounds of critiques. Unfortunately, we haven't received a response from the third reviewer despite our attempts to follow up. To prevent further delay in the peer-review process, and given that the two other reviewers, as well as I, have had the opportunity to review your revised manuscript and response letter, I believe I am able to make a fair judgment regarding your manuscript. Reviewer #1 has expressed a few minor concerns, however, I concur with Reviewer #2 about the scientific merit of your work and its potential impact. I also agree with both reviewers that your revisions have adequately addressed all major critiques. In order to avoid further delay, I recommend accepting your revised manuscript, contingent upon addressing the concerns raised by Reviewer #1 directly with the editorial team..

Reviewers' comments:

Reviewer's Responses to Questions

**Comments to the Author**

1. If the authors have adequately addressed your comments raised in a previous round of review and you feel that this manuscript is now acceptable for publication, you may indicate that here to bypass the “Comments to the Author” section, enter your conflict of interest statement in the “Confidential to Editor” section, and submit your "Accept" recommendation.

Reviewer #1: All comments have been addressed

Reviewer #3: All comments have been addressed

2. Is the manuscript technically sound, and do the data support the conclusions?

Reviewer #1: Yes

Reviewer #3: Yes

3. Has the statistical analysis been performed appropriately and rigorously? 

Reviewer #1: Yes

Reviewer #3: Yes

4. Have the authors made all data underlying the findings in their manuscript fully available?

Reviewer #1: Yes

Reviewer #3: Yes

5. Is the manuscript presented in an intelligible fashion and written in standard English?

Reviewer #1: Yes

Reviewer #3: Yes

6. Review Comments to the Author

Reviewer #1: I think the paper has improved a lot. I do have my doubts however, about the impact this work will. I wonder to what extent far reaching conclusions can be drawn about missing data in such surveys on the basis of just one dataset. However, I’ll leave it up to the editorial board to decide whether the impact is high enough in order for the paper to be publishable. One small textual comment: p. 1, lines 1-2 of second paragraph: “Since health surveys rely on voluntary participation”: is there any research where participation is not voluntary? Please, reformulate.

Reviewer #3: Although I wasn't one of the original reviewers, I was asked to weigh in on the paper. In doing so, I also had a look at the original reviews and the responses.

Overall, I think the paper is excellent and I don't really have any additional comments. The only thing that came up for me was an initial query around the negative binomial regression. In the main text, it's clearly indicated that the "outcome" is a count and that the negative binomial has (reasonably in my view) be adopted to acknowledge the potential for overdispersion. Nevertheless, this isn't clear in the Abstract (where the outcome is "missingness" and my initial thought was "why not a logistic model?"). With that, my only concrete suggestion would be to amend the abstract to make it clearer.

7. PLOS authors have the option to publish the peer review history of their article (what does this mean?). If published, this will include your full peer review and any attached files.

Reviewer #1: No

Reviewer #3: **Yes: **Sebastien Haneuse

---

## [Editor Report · Acceptance letter]

10 May 2023

PONE-D-22-22898R1 

Importance of missingness in baseline variables: a case study of the *All of Us* Research Program 

Dear Dr. Cronin:

I'm pleased to inform you that your manuscript has been deemed suitable for publication in PLOS ONE. Congratulations! Your manuscript is now with our production department. 

Kind regards, 

on behalf of

Dr. Bijan Najafi 

Academic Editor

PLOS ONE